# SLAV-Sim: A Framework for Self-Learning Autonomous Vehicle Simulation

**DOI:** 10.3390/s23208649

**Published:** 2023-10-23

**Authors:** Jacob Crewe, Aditya Humnabadkar, Yonghuai Liu, Amr Ahmed, Ardhendu Behera

**Affiliations:** Department of Computer Science, Edge Hill University, Ormskirk L39 4QP, UK; jacobcrewe@hotmail.co.uk (J.C.); humnabaa@edgehill.ac.uk (A.H.); liuyo@edgehill.ac.uk (Y.L.); ahmeda@edgehill.ac.uk (A.A.)

**Keywords:** simulators, reinforcement learning, autonomous vehicle, vehicle behaviour modelling, testing platform

## Abstract

With the advent of autonomous vehicles, sensors and algorithm testing have become crucial parts of the autonomous vehicle development cycle. Having access to real-world sensors and vehicles is a dream for researchers and small-scale original equipment manufacturers (OEMs) due to the software and hardware development life-cycle duration and high costs. Therefore, simulator-based virtual testing has gained traction over the years as the preferred testing method due to its low cost, efficiency, and effectiveness in executing a wide range of testing scenarios. Companies like ANSYS and NVIDIA have come up with robust simulators, and open-source simulators such as CARLA have also populated the market. However, there is a lack of lightweight and simple simulators catering to specific test cases. In this paper, we introduce the SLAV-Sim, a lightweight simulator that specifically trains the behaviour of a self-learning autonomous vehicle. This simulator has been created using the Unity engine and provides an end-to-end virtual testing framework for different reinforcement learning (RL) algorithms in a variety of scenarios using camera sensors and raycasts.

## 1. Introduction

The field of autonomous vehicles (AV) has been going through an innovation phase over the past few years. Companies like Waymo and Aurrigo are already rolling out commercial mobility services in the public domain, with other automotive companies like Tesla and Mercedes Benz planning to jump from semi-autonomous to fully autonomous driving [1,2]. However, deploying an autonomous vehicle in the real world requires a lot of testing of its software decision-making algorithms, sensors, and hardware components. For an autonomous vehicle to be perfectly functional in the real world, it must be thoroughly tested over many millions of miles, which is impractical in the real world [3]. Therefore, simulators play a key role in providing a low-cost virtual framework for testing autonomous driving systems [4].

Simulators usually consist of virtual representations of road networks, sensors, dynamic traffic situations, weather conditions, and vehicles where algorithms can be tested and rectified for a variety of factors exhibited in test cases. Simulation is also safer than real road testing, as it prevents accidents that can occur due to other vehicles and unique test cases that would be dangerous in the real world (e.g., zigzag roads with extreme weather, jaywalking pedestrians) [5]. Various complex driving scenarios can be easily generated, and developers can easily debug the software and hardware components based on the performance of the vehicle in the simulator.

Modelling vehicle behaviour has been one of the key components in not only improving the functionality of AV but also validating its own sensor networks in various road networks, traffic situations, and weather conditions. Having robust algorithms that can predict future states and make decisions in complex driving scenarios has been one of the key priorities in the AV industry. In recent years, reinforcement learning algorithms have been widely used to train and validate AV. Therefore, these algorithms are being used along with many simulators such as CARLA, CarMaker, etc. [6,7]. The key disadvantage of these simulators is that they are big pieces of software that involve having access to proper computation resources. Although these simulators are quite useful if there is a need to do a vast variety of experiments, it is better to have application-specific simulators that are lighter in size and can be used to perform in-depth validation linked to specific test case scenarios.

In this paper, we introduce a computationally efficient, easy-to-use self-learning autonomous vehicle simulator (SLAV-Sim) developed in the Unity game engine using Unity’s ML Agents. This simulator allows the user to develop various road scenes and scenarios and then test AV behaviour in these scenarios and assess the sensor performance of the camera and proximity sensors (using raycasts) in the vehicle. By utilising the built-in raycasting functionality provided by the Unity engine, users can perform ray-based collision detection and interaction with objects in a scene. This method takes advantage of highly optimised algorithms and data structures, resulting in fast and accurate raycast calculations. The raycast acts as a distance sensor (e.g., ultrasonic waves, IR, LED, etc.), and in combination with the camera, the framework is appropriate for the analysis and understanding of a given traffic scenario [8]. This allows for efficient collision detection and response, making it a computationally efficient technique for AV simulation [9]. The simulator also employs four reinforcement learning (RL) algorithms, namely the actor-critic method [10], deep deterministic policy gradient (DDPG) [11], deep Q-learning [12], and proximal policy optimisation (PPO) [13] to compare the effectiveness of the given AV in various road networks and traffic situations. Overall, the simulator generates rapid results with very little computational overhead for use in small-to-medium-sized research projects that aim to implement and test advancements in RL and CV rapidly.

The rest of the paper is organised as follows: In Section 2, we touch upon the different types of simulators available on the market, as well as the different RL algorithms that are used in the context of AV path planning. In Section 3, Section 4 and Section 5, we define the different components present in the SLAV-Sim simulator, the algorithms used, and the training setup, respectively. We then provide the reward and punishment metrics for modelling the vehicle behaviour in Section 6 and finally discuss the obtained results in Section 7.

## 2. Related Work

Situational awareness is an essential concept for AVs to safely navigate their surroundings. An AV needs to understand and anticipate the current and future states of its environment, including the presence of other vehicles, pedestrians, road signs, road conditions, weather situations, and potential obstacles [14]. Being able to understand and predict other road users’ behaviour using a camera would be beneficial to any AV [15,16]. However, current computer vision systems cannot yet achieve error rates acceptable for autonomous navigation [17]. Due to this, many researchers seek to combine the accuracy that can be obtained from a camera with other forms of detection, creating multisensor systems that can track several objects simultaneously [18]. These methods of detecting the environment must be dependable and accurate in all conditions in order to ensure safety. This need for safety leads to the research benchmarking of the reliability of sensors in different conditions, mostly camera systems, as these are the most likely to fail in adverse conditions [19,20]. Therefore, AVs rely on a combination of sensors, such as cameras, LiDAR, RADAR, and GPS, as well as advanced software and machine learning algorithms, to achieve situational awareness. These technologies work together to gather and process data about the AV’s surroundings, identify potential hazards, and make real-time decisions about how to navigate the road safely [21]. Maintaining situational awareness is crucial for AVs to avoid accidents and ensure passenger safety. However, it remains an ongoing challenge for developers and engineers to improve the accuracy and reliability of AVs’ sensing and decision-making capabilities, especially in complex and dynamic environments [22].

The need for AVs to have good situational awareness has led to many research papers that intend to map the immediate environment around a vehicle, recognise hazards, and plan paths to avoid accidents. Once an environment is observed, an AV must then use these observations to make predictions on what other road users will do and then react to these predictions and observations. An AV needs to use both a fixed and preplanned route as well as react to the immediate obstacles to get to its destination [17,23]. Taking action is the final necessary subsystem of an AV. These systems are far from ready for public use and require substantial development and testing. This testing can be conducted in the real world, through computer simulations, or a combination of both.

Testing is a crucial part of any system’s development, but this need is increased significantly when developing an AV. This field needs rapid and continuous testing in many scenarios with many different configurations to evaluate the robustness of a proposed system. In most modern methods of developing fully AVs, machine learning (ML) is actively used as a perception and decision-making component [24]. The nature of ML makes these systems difficult to debug, as an issue with any software will only arise if the AV is placed in such a fail-case scenario. Research into checking these ML components often utilises some form of simulation [25]. Due to these complications, simulation is quickly becoming a critical technique for testing different associated components. Since autonomous driving is a complex process that involves the collaborative working of different hardware, sensors, human-in-loop, and software components, it is crucial for the current simulators to provide a framework that tests and tunes vehicle behaviour in different scenarios [24,26].

The current market consists of various simulators, each designed to cater to a different purpose. For simulators involving general robotics, Gazebo is a popular simulation platform that provides a framework to accommodate different sensor models and physics engines to be plugged in [27]. Although it is useful for simple robotics simulations, its inability to run the current AVs demands complex environments and test cases. Commercial automotive simulators such as ANSYS, dSPACE, PreSCAN, rFpro, Cognata, and NVIDIA’s Drive Constellation dominate the market space with their wide variety of test case generation, sensor configuration, and testing features [28,29,30]. However, these simulators are not open source. For a brief period of time, video games such as Grand Theft Auto V (GTA 5) were used to generate test cases for research [31]. However, the complexity of accessing different components of the game due to user licence issues and the lack of sensor support made this technique implausible.

Unreal and Unity game engines are the large open-source game engines that have been gaining traction in simulating AVs in recent years [32,33]. Although these engines usually cater for game development, simulators based on these open-source platforms, such as CARLA [7], DeepDrive [34], and AirSim [35] have been predominantly used for AV research. These simulators allow users to synthetically generate driving scenes and test-case scenarios to test ML algorithms using data from various sensors. The only disadvantage of these simulators is that they have large software sizes and require heavy computation, which can be detrimental if a user wants to test only on specific test cases and conditions. Application-specific simulators are being developed to cater to specific test cases and applications in the wider AV domain. AutonoVi-Sim involves the rapid development and testing of autonomous vehicle simulators in complex environments with different weather conditions [36]. ViSTA provides a test-case generation framework for the performance evaluation of AVs based on sensor parameters [37]. ViSTA 2.0 integrates this test-case generation with virtual environment development, sensor configuration, and evaluation capabilities [38]. A high-fidelity simulator called LGSVL provides end-to-end simulation with a sensor customization function [39].

Reinforcement learning is useful for training autonomous agents to make optimal decisions in complex and uncertain environments, enabling applications. The traditional implementation of SAC, PPO, DDPG, and DeepQ have often been used in human actions [10,11,12] and game logic [12,13]. Lately, these algorithms have been adapted to be used in simulation-based vehicle testing. RL algorithms allow AVs to learn from their environment and make decisions based on the outcomes of those decisions [40]. In the context of autonomous driving, RL can be used to teach an AV how to navigate complex and dynamic environments by rewarding it for making good decisions and penalising it for making bad ones. For example, an AV could be rewarded for successfully avoiding a pedestrian or for taking a safe and efficient route to its destination [41]. One real-world example of the use of RL in autonomous driving control is Waymo’s self-driving taxi service. Waymo uses RL algorithms to train its autonomous vehicles to make safe and efficient driving decisions in a variety of situations. For example, the vehicles learn how to handle complex intersections and navigate around parked cars and other obstacles [42]. Traditional RL algorithms such as deep Q-network (DQN) and deep deterministic actor–critic (DDAC) algorithms are actively used to predict the proper steering angle, acceleration, and brake values for the lane-keeping function [10,12,13,40]. The policy gradient algorithm (PPO) has shown high effectiveness in multiagent settings despite the belief that it is significantly less efficient than off-policy methods in the context of multiagent systems [43]. This has made it an active algorithm to be directly used in decision-making for AVs. An approach by Shwartz et al. uses PPO to perform collision-free path planning for the agent [44]. The double deep Q-network (DDQN), which uses visual representations as system inputs, was used by Zhang et al. to construct a vehicle speed control technique [45]. For motion control in a roundabout scenario, Chen et al. studied four model-free DRL algorithms: DDQN, deep deterministic policy gradient (DDPG), twin delayed DDPG (TD3), and soft actor–critic (SAC) [46]. The results showed that the SAC technique outperformed the others. In order to provide positive and reliable performance, DRL has also been included in specific motion control modules on the vehicle. Chae et al. utilised DQN with a carefully crafted reward function for an adaptive braking system to prevent crashes [47]. To improve the performance of MPC controllers, Ure et al. successfully implemented DRL to alter the settings [48]. For navigating around obstacles, Zhou et al. employed DDPG, which first teaches the agent to navigate a scene without barriers before adding prospective impediments and defining the agent’s control strategy [49].

Liu et al. introduced a novel approach by incorporating a modified policy network into the SAC structure to address the dual objectives of increasing reward and emulating expert behaviour. Their algorithm was evaluated on a simulated roundabout scenario with dense traffic, demonstrating a higher success rate compared to baseline algorithms [50]. In order to assess the effectiveness of RL algorithms in real-world scenarios and compare them to simulated approaches, Feng et al. proposed the dense deep-reinforcement-learning (D2RL) method. It leveraged densified information, including safety-critical events, to tackle tasks that traditional deep RL approaches found challenging. The results indicated that D2RL was an advanced technique capable of accelerating the autonomous vehicle testing process in both real and virtual environments [51]. These studies highlight the efficacy of innovative algorithmic enhancements, such as modified policy networks and the utilization of densified information, in improving the performance and applicability of RL algorithms. Both approaches offer promising advancements in addressing complex scenarios and bridging the gap between simulation and real-world testing of AVs.

## 3. Scenario

The SLAV-Sim simulator uses the open-source Unity engine and its ML agent to develop various driving scenarios and train the agents. Concerning resource management, simulators using Unity ML Agents demonstrate superior resource efficiency in comparison to others that have used ML packages such as Unreal Engine, TORCS, and CARLA. Unreal Engine is renowned for its robust rendering capabilities but often demands considerable resources due to its sophisticated physics systems [52,53]. On the contrary, Unity and its ML Agents optimise resource usage through techniques like batch processing, parallelization, and memory management, prioritizing efficiency even on modest hardware [54].

For example, Santara et al. have developed a multiagent driving simulator (MaDRAS), which is built upon TORCS. However, it does have its own limitations—it may not match up to some commercial simulators in terms of realism, graphics, and physics. It also has a limited set of environments and tracks, potentially constraining the diversity of research scenarios. Integrating the MaDRAS simulator with external systems can be challenging, impacting interoperability. Additionally, it lacks some advanced machine learning capabilities. Moreover, TORCS does not fully capture real-world variability and offers less customization and flexibility compared to more advanced simulators. Lastly, community size and update frequency may affect its ongoing development and feature enhancement [24]. The CARLA simulator is known for its realistic autonomous driving simulations, often requiring substantial resources to maintain their fidelity. It is unable to handle a dynamic simulation environment and has to rely on manual code and multiple conditional statements to simulate any scenarios [26,55,56]. Unity strikes a balance by providing a lightweight and efficient simulation experience via a dedicated ML agent framework without compromising on realism [24,57]. Therefore, SLAV-Sim gains a competitive edge over other simulators by ingeniously utilizing a more compact Unity package. This strategic choice translates into superior efficiency and accessibility, as the simulator optimizes its performance without sacrificing essential features. The smaller package ensures a swift setup, reduced memory footprint, and quicker downloads, making it an ideal option for researchers and practitioners, even in resource-constrained settings.

In SLAV-Sim, Unity holds the environment and its dynamics and gives rewards and punishments based on the actions of various agents. The observational data (images and raycasts) are then passed to ML Agents (TensorFlow), which convert them to the data representation that is used by various RL algorithms to estimate the next move or action. The observational data for move/action estimation are implemented using Python. The reward or punishment is then passed back to ML Agents and then through Unity, where the action or movement takes effect in the physics simulation and the new state is observed. Finally, this observation is passed to ML Agents and repeated until the required number of steps is passed. Figure 1 shows the overview of the SLAV-Sim simulator.

The SAC and PPO algorithms come as part of the ML Agents’ package, but the DDPG and deep Q algorithms use Open AI Gym [58], an API for running environments. This makes all the scenarios into separate Gym environments. ML Agents also come packaged with their own API to communicate with Gym, which means that any reinforcement learning algorithm can be used in this environment.

### 3.1. Road Network

All the virtual scenarios are created to imitate the real-world scenarios, primarily the United Kingdom Highway Code [59]. These environments are constructed in Unity to allow the scenarios to use Unity’s physics engine. This provides an accurate simulation of how objects would move with near-to-life realism and also provides access to all the Unity tools to speed up the production. The environment is split into different scenarios, with each scenario having a different road layout that represents a different part of driving. Each of these scenarios is in its own environment, which permits the AV to train for certain aspects of driving in separate spaces. This also allows for comparisons between each algorithm when controlling the vehicle in different scenarios. All the roads in each scenario have colliders extruding upward from the middle and sides of each road segment. These walls allow the AV to monitor its position relative to the edge and middle of the road and detect when the AV has gone off track. These walls are present on every piece of road in all the different scenarios. The vehicle can pass through the wall in the middle of the track without a collision. This wall affects the raycast observations, and the vehicle will be reset if it touches one of the outer walls. An example of the collider can be observed in Figure 2.

The basic driving, speeding, roundabout, and traffic scenarios, as the names suggest, test the AV’s behaviour in general driving, speeding, manoeuvring, and traffic conditions. The 4-way lights and crossing add stop lights as additional infrastructure, which tests the stopping control of the AV. These scenarios mimic city driving and are set to low-speed limits. The motorway is a high-speed limit scenario where the agent must learn to approach this limit and control the vehicle, keeping in lane at high speeds. The scenarios are detailed in Figure 3.

The AV is the simulated car that is controlled by the agent, which is driven by the RL algorithm. It has two components, namely, physics and perception. The physics and perception components are generated and manipulated using the Unity environment.

#### 3.1.1. Physics

The vehicle reacts to physics, and the agent passes action commands for any reaction. These commands are speed, acceleration, deceleration, and turning angle (left or right). SLAV-Sim uses Unity wheel colliders and a custom-made car controller to simulate the physics of the vehicle. The Unity wheel colliders have built-in collision detection, wheel physics, and slip-based tyre friction. These wheel colliders allow for realistic tyre physics in real-time simulation without adding to the time. When the agent accelerates, force is applied directly to the front two tyres; this allows the acceleration, braking, and turning to all be simulated accurately. The Agent’s ego view can be seen in Figure 4.

#### 3.1.2. Perception

As far as vehicle-to-everything (V2X) interactions are concerned, the agent is the only dynamic entity present in the scenario, and it interacts with the environment (the road network and signals) through camera and raycast sensors. The agent needs information from the environment to make decisions on what actions will lead to the best outcome. With real-world AVs, many different sensors would be used to map the surrounding world. However, our objective was to simulate the behaviour of an AV in a virtual scenario with the known positions of various static structures and dynamic objects (other road users). Thus, the distance from the target vehicle to the sides, middle of the roads, and other road users was computed using raycasts.

A raycast is a straight line from one point in 3D space to another. It can return the distance from the origin to the first object it hits and is similar to a LiDAR (light detection and ranging). The raycast walls are visualised in Figure 5. These raycast distances are passed to the RL algorithm as continuous observations. Raycasts in the Unity engine are computationally inexpensive due to several optimisation strategies. One of them is the deployment of acceleration structures like bounding volume hierarchy (BVH), allowing for the efficient organization of the scene’s geometry and rapid culling of irrelevant portions for ray–object intersection tests. By spatially partitioning the scene into manageable chunks, Unity significantly reduces the number of checks required during raycasts [60]. Moreover, optimized algorithms, caching mechanisms, and SIMD (single instruction, multiple data) optimization ensure that ray–object intersections are calculated with minimal computational overhead. Unity’s raycasting API is finely tuned to streamline the process further, incorporating early exit strategies when intersections are found, effectively maximizing computational efficiency during collision detection [61].

By observing the environment in this way, the agent can create a map of its environment with a small number of inputs to the RL algorithm. This improves the computational efficiency for handling the target AV at higher speeds and modelling complex scenarios. In SLAV-Sim, the agent receives observations consisting of ten raycast distances, the rotation, and the speed of the vehicle. Three raycasts are situated at the front, three raycasts on the left and right, and one raycast at the back of the agent. The raycast positions were chosen to reflect the importance of the information being shown. All the observations are passed as two stacked vectors. This allows the agent to work out the direction of travel by comparing the length of any raycast between the last observation and the next. The agent can also work out if it is drifting by comparing the rotation observations. Observing the environment is also possible using a camera (an egocentric view of the AV). This normally replaces the raycast-based distances with frame-by-frame image data to observe the environment. However, the image data are much larger than the raycast data for the RL algorithm, as each pixel in an image is an input data point. As a result, this creates a much larger model and more complex analysis, impacting real-time behaviour simulations. Thus, it is impractical to use high-resolution images as inputs to RL algorithms, as this slows down the training due to high computations as well as increasing inference time. In this framework, the image resolution of 20 × 20 was chosen experimentally without compromising the model’s accuracy. This image was converted to a high-level feature representation using a standard pretrained deep convolutional neural network such as IMPALA [62]. The IMPALA network fares better computationally than the traditional ResNet implementation [63]. The Unity machine learning agents recommend a resolution of 20 × 20, which is sufficient for real-time modelling and analysis.

## 4. Control and Algorithms

SLAV-Sim uses the stable baseline implementations of RL algorithms such as DDPG and DeepQ and the TensorFlow implementations of SAC and PPO [12]. All the algorithms use discrete actions except for DDPG, which uses continuous action spaces, and therefore the control of the vehicle is slightly different.

While using a discrete agent, there are two 1 × 3 arrays. The individual arrays in the context represent three distinct actions: braking, lane changing (discrete actions), and turning or acceleration (continuous actions). Therefore, the agent outputs two numbers, one of which represents the motor actions (forwards, backwards, and braking), and the other represents the steering input (left, straight, and right). In order to provide the agent with finer control over the steering angle, the selected inputs are smoothed, allowing for a gradual transition from a straight position to a full turn. This approach introduces a time delay for the wheels to change direction, enabling the agent to execute more precise turns by oscillating its turning action at each step when utilising discrete actions. The actions are visualised in Figure 6.

A continuous action space means that the agent will output a real number between −1 and 1. This means that a continuous agent can choose from many more actions than a discrete algorithm. Continuous actions allow for more control over the vehicle than discrete actions, which only allow binary inputs. It could slow down initial learning as a continuous agent has many more possible decisions that can be made, meaning that collecting initial rewards could be more difficult.

## 5. Experiment Design

This section discusses the experiment design involving the use of ML Agents for testing. The raycast features were fed into a 2-layer dense network for the final feature representation. The high-level features from the pretrained ResNet model along with the final raycast feature representation were then passed through the RL algorithms (DDPG, DeepQ, SAC, and PPO) for modelling the AV’s behaviour. Each experiment conducted enabled a comparison of the training performance and driving behaviour exhibited by each algorithm at different stages. The training process involved a maximum number of 10 million (M) steps. This step is divided into shorter segments (1 M, 2.5 M, and 5 M) for an intermediate analysis to evaluate the progress achieved at each training milestone. Each algorithm was trained once on each scenario, and the resulting models were saved for analysis using inference. By employing inference, the models no longer underwent training but instead aimed to exploit known rewards as frequently as possible. This approach provided a comprehensive performance assessment of all agents at each training milestone for all scenarios, enabling meaningful comparisons between them.

### 5.1. Description

A total of four experiments were conducted to assess the performance of ML Agents across different scenarios.

#### 5.1.1. Learning Schedule Experiment

In this experiment, the ML agent’s implementation of SAC and PPO was utilised, keeping the hyperparameters (e.g., learning rate, batch size, etc.) nearly unchanged from the basic agent setup provided by ML Agents. This setup yielded the desired behaviour and served as a benchmark for further improvements. The focus was on exploring the effect of the learning schedule, specifically the learning rate, which is responsible for guiding the agent’s learning process. A constant learning rate was chosen as it had no impact on the computing time and had a consistent impact on the performance.

#### 5.1.2. Additional Algorithms Experiment

To perform experiments using DeepQ and DDPG, each scenario was converted into an Open AI Gym environment. SAC, PPO, DeepQ, and DDPG were all compared. This experiment was conducted to show the usefulness of the simulator as a benchmarking tool for a wide variety of RL algorithms.

#### 5.1.3. Observation Experiment

It compared the performance of camera observations against raycast observations. The aim was to determine which observation method performed best in each scenario. This experiment used only PPO as it was determined to be the best algorithm in the additional algorithms experiment. The network size of the algorithm needed was increased for the camera agent due to passing additional inputs (images via ResNet). The network size of the raycast was also increased to ensure a fair comparison.

#### 5.1.4. Increased Network Size Experiment

The final experiment used the large PPO network. It used raycast data from the observation experiment and compared them to the small PPO network used in the additional algorithms experiment. This showed what performance improvement the larger network gave to this agent.

### 5.2. Training Metrics

A set of quantitative training metrics such as *cumulative reward* and *episode length* were utilised to allow a direct comparison between all agents.

#### 5.2.1. Cumulative Reward

The overall reward is updated at each step during training, representing the cumulative reward obtained by the agent. It is set to 0 initially and incremented or decremented as the agent received rewards or punishments, as discussed in Table 1. A positive cumulative reward indicates that the agent has received more rewards than punishments throughout the training process so far. These cumulative rewards are visualised using line graphs, where the steepness of the lines reflects the rate at which the algorithm accumulates rewards.

#### 5.2.2. Episode Length Line Graph

This metric shows crash frequency during training by measuring the length of every episode. An episode is the training performed before the agent needs to be reset. This graph shows how long the agent lasts before crashing during training and is used along with the cumulative reward to calculate how well each algorithm did during training.

### 5.3. Testing Metrics

Testing used the brains saved at 1 M, 2.5 M, 5 M, and 10 M steps. A crash location image is a qualitative image of the locations where agents crashed. One image is created per brain, and this is used to locate where the agents crashed most frequently. This metric was chosen to allow for the analysis of situations in which agents were crashing. This information was used to work out the most difficult sections of a scenario. These images were also used as a method to show improvement at the different training stages, as a better agent would typically crash less. Videos of the agents were taken to show their behaviour during testing to allow for comments and analysis of the behaviours learned by each agent and as a method to observe this behaviour for binaural anomalies.

### 5.4. Hyperparameters

For the raycast algorithms, the hidden unit was set to 128 and the number of layers to 2 for all algorithms. The batch size was kept at 2048, and the number of steps was kept at a maximum of 10 M. The learning rate for these experiments was set at 3.0 × 10^−4^. These values are set experimentally.

### 5.5. Crash Frequency

The crash frequency bar charts were generated by monitoring the length between crashes during testing. This was done at the same time as the crash frequency report. These values were then averaged out to show the overall crash rate of each agent at each stage of training. The details about the crash frequency and the crash frequency report are present in the Appendix A.

## 6. Rewards and Punishments

A set of rewards and punishments were introduced to train the autonomous vehicle to successfully follow a path. The rewards and punishments values were based on the UK highway code [59], which provides guidelines for safe and responsible road use in the United Kingdom. Some of the rewards scaled and increased in value as the vehicle learned to navigate the environment. High reward and punishment values were provided for safety-critical actions such as *traffic light action* and *crash*. Table 1 provides an insight into the observations made through a reward and punishment chart. SLAV-Sim allows for a modification of rewards and punishment metrics based on the preliminary results as well as the inclusion of new scenarios as evidenced by work done by similar simulators [64]. A descriptive description of the rewards and punishments is provided in the Appendix A.

## 7. Results and Discussion

### 7.1. Learning Schedule Results and Comparison

The SAC and PPO were compared using constant and linear learning rates. This is shown in Figure 7. A linear learning rate reduces the learning rate as steps are completed; this makes policy updates smaller when approaching the perceived optimal policy. A constant learning rate stays the same throughout training. The cumulative reward graph showed that the PPO constant performed slightly better than the SAC constant. The SAC linear and PPO linear had lower performance. Initially, all RL algorithms had a gradual increase in cumulative reward, but around 3M steps, they experienced a significant gain in rewards per step. All RL algorithms demonstrated the required driving behaviour in all scenarios without receiving negative rewards at the end of training. These results confirmed the correctness of the environment and training setup. The constant learning rate outperformed the linear one in both algorithms, making it the preferred choice for future experiments.

### 7.2. Additional Algorithm Results

DDPG and DeepQ were selected as the additional algorithms to test along with the traditional algorithms. The graphs in Figure 8 show an example where algorithm performance is measured on a specific scenario, whereas Figure 9 shows the cumulative reward on all scenarios. PPO outperformed the other algorithms in terms of cumulative reward, achieving higher scores in five out of seven scenarios. The driving behaviour generated by PPO aligned with expectations. Despite experiencing more crashes in the episode-length experiments, PPO drove closer to the speed limit compared to others. This allowed PPO to accumulate rewards by driving faster and receiving more forward rewards. However, the higher speed led to more frequent crashes and affected performance in the crossing scenario, where stopping at crossings was crucial. PPO initially prioritised forward rewards but improved when it discovered the value of waiting at crossings. SAC, although slower in speed compared to PPO, demonstrated better control and fewer crashes.

However, the slow speed resulted in missed opportunities for substantial forward rewards and lower cumulative rewards. SAC showed a desired driving behaviour earlier in training for five out of seven scenarios, indicating its sample efficiency. DeepQ showed desirable behaviour but did not achieve rewards as high as SAC or PPO. It had longer episode lengths, lower speeds, and slower improvements in rewards. The algorithm missed out on potential rewards similar to SAC due to its slow speed. Comparing these algorithms validated the correctness of the reward and punishment settings across multiple approaches and confirmed the reliability of the simulator. It also provided insights into challenging learning behaviours and highlighted the significance of specific reward functions, such as the importance of forward rewards for PPO.

### 7.3. Observation Methods

As shown in Figure 10 and Figure 11, the camera observation method achieved the highest rewards in two (basic driving and motorway) of the seven scenarios. The large raycast network achieved the highest reward in the other five scenarios. The small network raycast agent outperformed the camera agent when accurate distance measurements were required in a scenario.

The results clearly indicate a positive correlation between network size and performance, as the larger network raycast model consistently outperforms the smaller network in every scenario. This finding suggests that even with a limited number of inputs, the agent can effectively gather sufficient information to accurately map its surroundings. Comparing the performance across different scenarios for each observation method, it reveals that the camera observation method excels over the raycast method in scenarios where precise distance measurements are not critical. For instance, in the motorway scenario, where the camera outperforms the raycast, the camera’s primary task is to identify the road edges and centre.

The camera agent demonstrates superior performance in scenarios that require straightforward driving behaviour, thanks to its access to more comprehensive information about the upcoming track. This advantage arises from the camera observation method’s focus on the forward direction, providing the agent with detailed information about the road ahead. Consequently, the camera agent excels in certain scenarios where its ability to gather detailed forward-looking data is beneficial. However, the camera agent faces challenges in scenarios such as crossings, where it is necessary to come to a complete stop before reaching the edge of the crossing. The camera agent’s lack of accurate distance information often results in failures to stop in a timely manner. This limitation also affects the agent’s responsiveness to other vehicles in the traffic scenario. Despite its strengths in certain scenarios, the camera agent’s inability to accurately measure distances hinders its performance in situations that require precise spatial awareness and interaction with the environment.

## 8. Traffic Flow Integration

An exciting potential enhancement for SLAV-Sim involves the integration of real-world human-driven traffic demand, thereby simulating authentic background traffic flows for testing AVs. This integration can be achieved in Unity by incorporating traffic flow information. AI can be programmed to emulate the behaviour of the traffic flow information, considering factors such as traffic rules, driving styles, and responses to various road conditions. By incorporating traffic flow information that dictates realistic behaviours for these agents, a complex network of human-driven traffic can be generated [65,66]. Unity’s physics engine and NavMesh system can also aid in managing the movement and interaction of these agents, simulating real-world traffic scenarios, and providing a robust environment for the comprehensive testing and validation of AVs in diverse and dynamic traffic conditions [67]. Additionally, the NavMesh system can also be used by SLAV-Sim to incorporate complex scenarios such as advanced car-following and lane-changing behaviours by dictating how vehicles follow one another, adjusting speeds, and performing lane changes realistically [68,69]. SLAV-Sim can easily integrate these AI-driven algorithms to simulate intricate traffic dynamics, enhancing its capability for comprehensive testing of AVs in varied and challenging traffic scenarios.

## 9. Computational Resources

All the settings in our simulator were tuned to run on the lowest of the specifications. We tested our simulator on a low-spec (two-core, four-thread Intel i3) laptop as well as a midrange (AMD 2700x with 8 cores and 16 threads) CPU. The training time for these specifications was 3.5 and 2 h, respectively, for the 10 million steps. This was around 1.3 and 0.7 milliseconds per step, respectively. All the settings in our simulator were tuned to run on the lowest of the specifications, justifying the lightweight nature of the simulator. We suggest other approaches that can be incorporated to improve the speed of SLAV-Sim simulations for testing connected and autonomous vehicles (CAV). Research done by Matija et al. [70] utilizes level-of-detail (LOD) techniques in their automotive simulations, which strategically reduce the detail of objects as they move farther from the camera. This enhances their rendering performance without compromising on visual fidelity or simulation speed. Asset optimization can be another technique where the polygon count and the asset size of the 3D models can be minimized while maintaining visual quality. Paranjape et al. [71] used techniques like texture atlases to reduce memory usage and improve loading times for the quick construction of digital twins. Additionally, implementing culling and occlusion techniques, such as frustum culling and occlusion culling [72], can help in excluding objects that are not visible to the camera, saving valuable rendering resources and further enhancing performance. To achieve flexibility in test scenarios without affecting frame rates, simulation time scaling can be implemented. This feature allows for the acceleration or deceleration of the simulation time while maintaining the desired frame rate, enabling a wide range of testing conditions. Multithreading is another critical strategy that can be utilized to enhance performance by distributing computation-intensive tasks across multiple threads and taking advantage of modern multicore processors. Efficient asset loading and management can be achieved through streaming and asynchronous loading, dynamically loading assets as needed, and minimizing loading times. GPU performance optimization and real-time physics optimization are essential for leveraging the GPU’s capabilities effectively and adjusting physics simulations based on object distance and importance within the scene, respectively.

## 10. V2X Integration

Currently, SLAV-Sim primarily focuses on simulating the behaviour of an autonomous vehicle concerning its environment. However, the simulator provides a potential platform for others to adapt it according to their requirements, including the integration of V2X (vehicle-to-everything) technology. Incorporating V2X within SLAV-Sim entails extending the simulation to model V2X communication protocols and interactions. Given Unity’s capabilities, this integration can be relatively straightforward. Our simulator offers a robust environment for building complex networked systems, making it well-suited for implementing V2X communication frameworks [73]. Leveraging its networking capabilities, protocols like DSRC (dedicated short-range communications) and C-V2X (cellular vehicle-to-everything) can be emulated within the simulator. Behavioural metrics can be created by closely simulating real-world driving behaviours within SLAV-Sim. Users can import data containing vehicle trajectories and behaviour information and then simulate them, considering parameters such as speed, lane adherence, and acceleration profiles. These simulated behaviours can then be compared with real-world vehicle trajectories captured from similar scenarios to evaluate alignment. Similar work done by Tang et al. [74] and Fremot et al. [72] used quantitative metrics such as mean absolute error (MAE) and root-mean-square error (RMSE) to measure discrepancies in positions, speeds, and other parameters. Additionally, lane adherence scores are calculated to assess how well simulated vehicles adhere to real-world lane patterns. SLAV-Sim also has an inbuilt graph maker that can be utilized to create visualizations like scatter plots and line charts for a clear comparison. Communication metrics can be calculated by integrating V2X communication protocols within the SLAV-Sim. Message latency can be measured by tracking the time taken for V2X messages to propagate from sender to receiver in any simulation. The message loss rate can be determined by comparing the number of sent messages to the number of received messages. Similar work has been done by Wang et al. [12] and Oczko et al. [75], where V2X communication protocols were integrated using appropriate networking libraries. These V2X communication values were then compared with the recent European V2X trials for equivalency. Safety metrics, including collision rates and near misses, can be created by implementing collision detection algorithms within the SLAV-Sim simulation. The frequency and type of collisions and near-misses can be recorded and analysed for comparison with real-world safety data. By comparing these metrics with real-world safety data from similar scenarios, the simulation’s accuracy and alignment with real-world safety standards can be evaluated effectively. By extending SLAV-Sim to encompass V2X, researchers can delve into comprehensive studies and experiments related to V2X technology seamlessly within this flexible and powerful simulation environment [76,77].

## 11. Conclusions

In this paper, we revisited existing AV simulation tools and addressed the need for a more accessible solution. Our objective was to develop a lightweight and capable simulator using Unity ML Agents, an open-source tool, for the effective benchmarking of RL algorithms. Using Unity ML Agents, we compared the performance of four RL algorithms. The PPO algorithm emerged as the most effective among the tested agents, while continuous actions proved less successful within the allocated training time. Expanding on our findings with the PPO, we explored the impact of observation methods using a camera to observe the environment. We compared two observation methods and investigated the effect of increasing the algorithm network size. This analysis provided valuable insights into performance changes between algorithms with the same observation type. In conclusion, our research presents a computationally efficient and capable AV simulator. We established the superiority of the PPO algorithm and gained insights into the impact of observation methods and network size on performance. Our accessible simulator, with its benchmarking capabilities, serves as a valuable tool for advancing autonomous vehicle research.

## Figures and Tables

**Figure 1 sensors-23-08649-f001:**
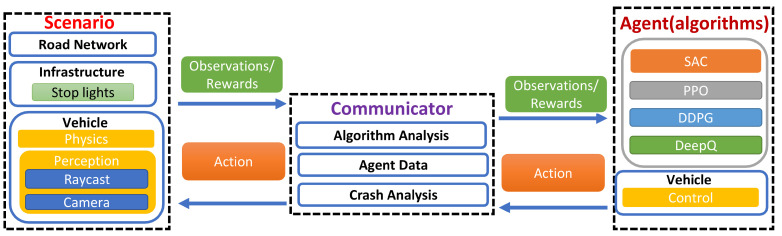
Platform overview of the SLAV-Sim simulator: The simulator is divided into eight modules, each focusing on a different aspect of autonomous driving. The driving environment is defined using the road network and infrastructure. The vehicle component consists of vehicle physics and perception modules. Different combinations of these components give rise to multiple scenarios in the Unity environment. The agent consists of the different algorithms that are used for training the vehicle as well as basic control strategies. Unity ML Agents act as a communicator between these two sections and provide services such as algorithm and crash analysis as well as access to agent data.

**Figure 2 sensors-23-08649-f002:**
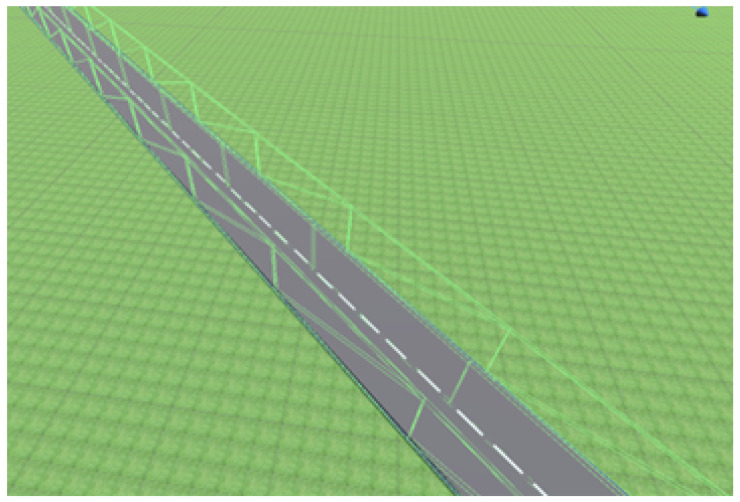
Road example showing the raycast walls in green boxes.

**Figure 3 sensors-23-08649-f003:**
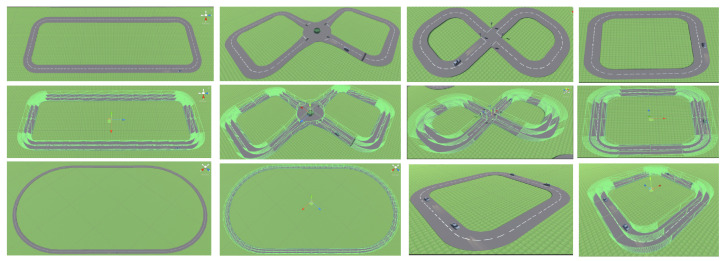
The diagram showcases different driving scenarios, presenting them with and without the inclusion of raycasts. The first row depicts speeding, roundabout, 4-way lights, and crossings without the utilization of raycasts. In the second row, the same scenarios are displayed, but this time incorporating raycasts. Lastly, the third-row features motorway and traffic scenarios, illustrating both cases with and without raycasts.

**Figure 4 sensors-23-08649-f004:**
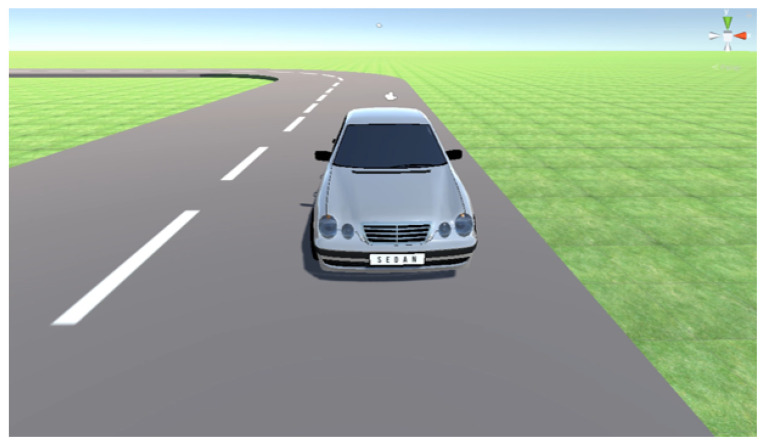
Agent’s EgoView.

**Figure 5 sensors-23-08649-f005:**
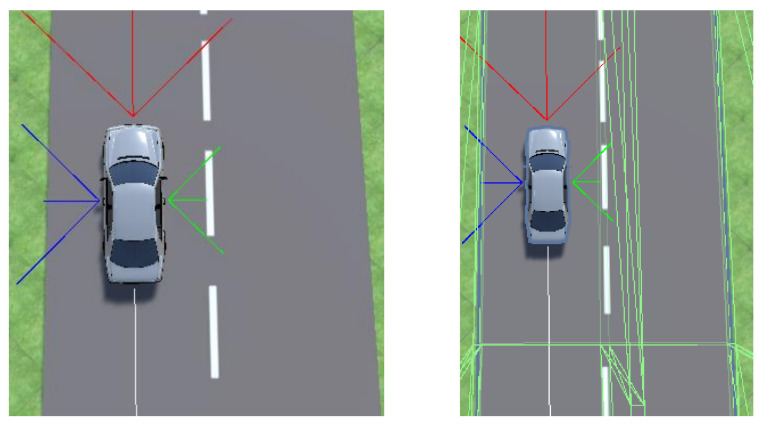
Raycasts without invisible walls (**left**) and with invisible walls (**right**).

**Figure 6 sensors-23-08649-f006:**
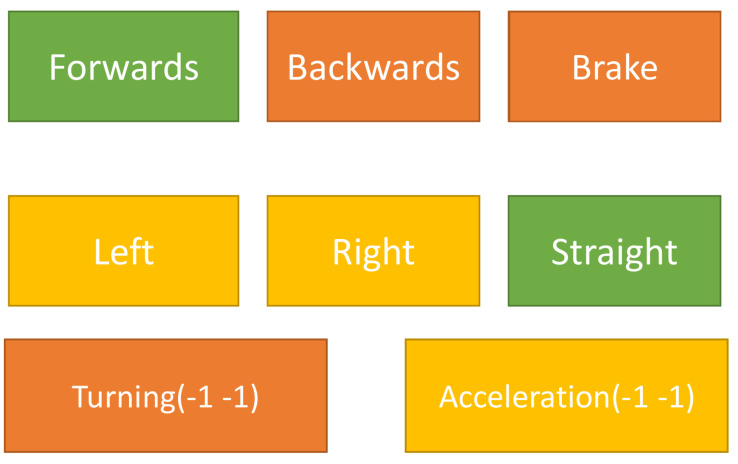
This diagram represents all of the actions available to the discrete agent. The agent can choose one action from each row at every step. The continuous agent number chooses between two or more actions from the discrete algorithm.

**Figure 7 sensors-23-08649-f007:**
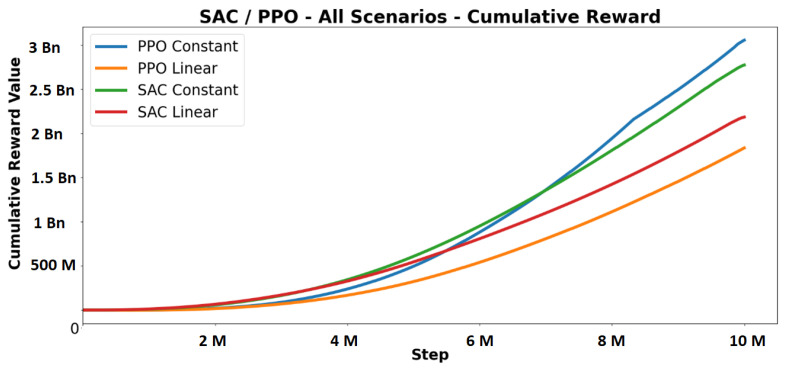
Baseline results for SAC and PPO algorithms on all scenarios. The PPO constant agent performs slightly better than the SAC constant agent in the long term.

**Figure 8 sensors-23-08649-f008:**
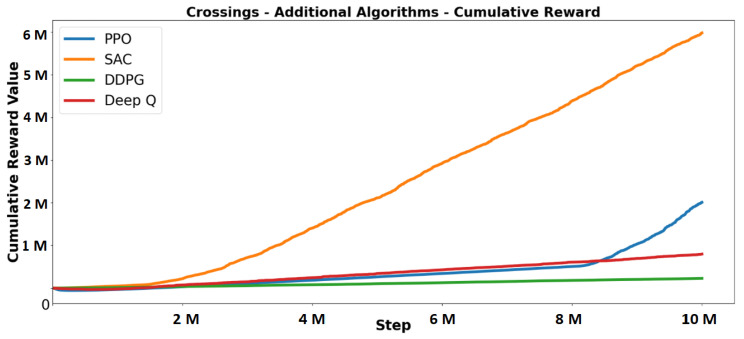
Crossings’ cumulative reward and episode length.

**Figure 9 sensors-23-08649-f009:**
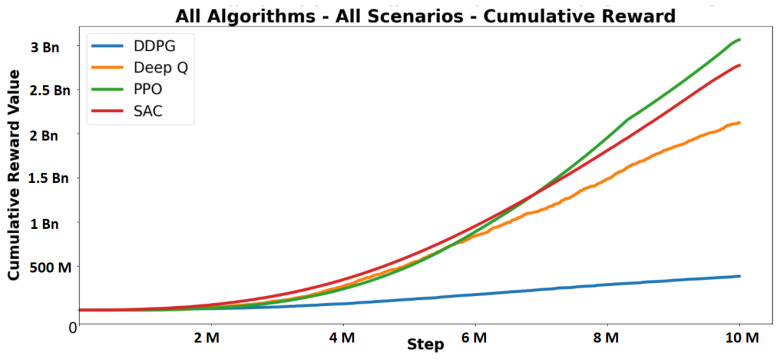
Cumulative reward for all scenarios.

**Figure 10 sensors-23-08649-f010:**
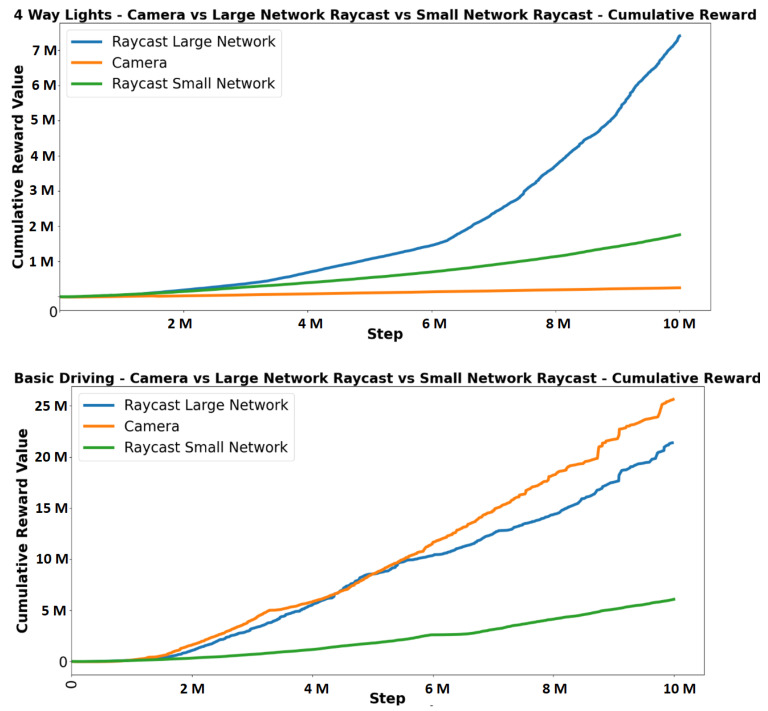
Comparison of cumulative rewards between camera, large network raycast and small network raycast for 4-way lights and basic driving scenarios.

**Figure 11 sensors-23-08649-f011:**
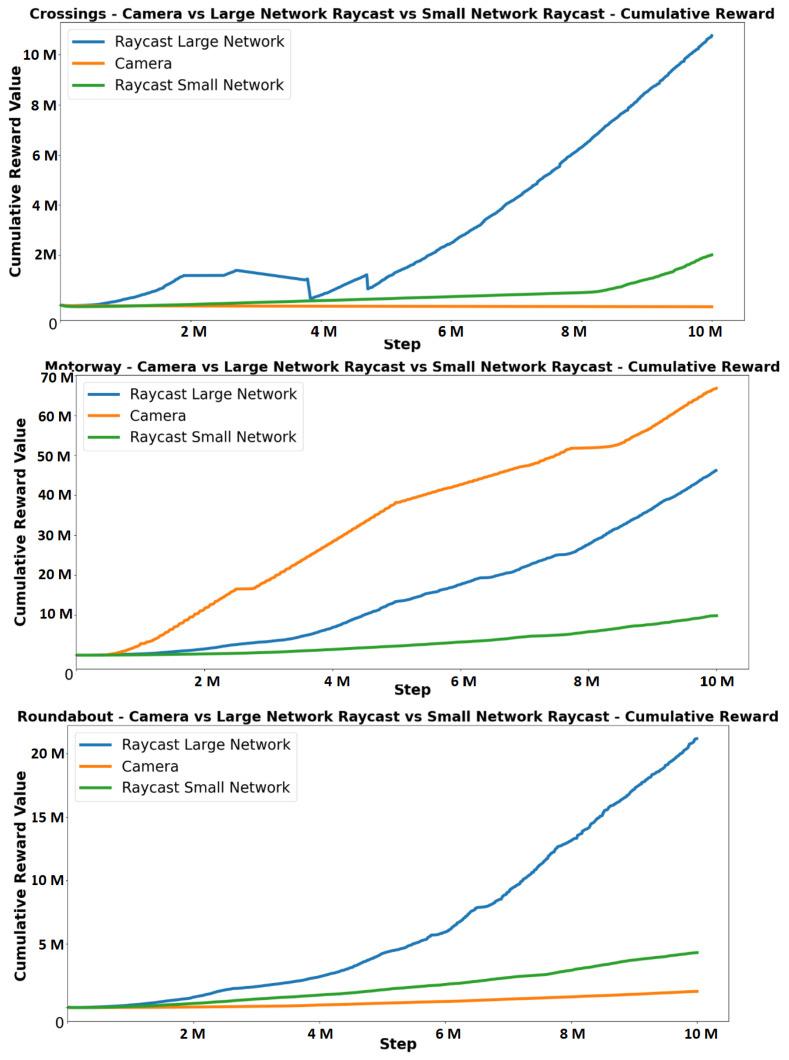
Comparison of cumulative rewards between camera, large network raycast and small network raycast for crossings, motorway and roundabout scenarios.

**Table 1 sensors-23-08649-t001:** Rewards and punishments chart.

Observation	Rewards	Punishment	Comment
Correct side of the road	+0.1		The agent is encouraged to consistently remain on the correct side of the road.
Middle of the road		−0.5	Encourages the agent to stay between the wall and middle line of the road
Moving forward	Reward Value = (Speed limit − Distance from speed limit)/10	−0.5 (Above speed limit)	Prevents backward driving during initial learning
Correct braking	+0.5		Encourages the agent to brake before crashing.
Unnecessary braking		−0.5	Promotes a smoother driving
Distance from the wall		−0.02 (>2 mtr) −0.05 (<1 mtr)	Prevents chances of crashing by encouraging the vehicle to stay in the lane
Traffic light pass when green	+10		Promotes traffic light behaviour understanding
Stop zone at red light	+0.15		Promotes traffic light behaviour understanding
Crash		−15	Prevents vehicle from going off-track

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
