# Peer review of "SLAV-Sim: A Framework for Self-Learning Autonomous Vehicle Simulation"

_sensors, 2023, doi:10.3390/s23208649_

Round 1

Reviewer 1 Report

The paper stands as a laudable endeavor, shedding light on an innovative autonomous vehicle simulation framework and pioneering a benchmark for RL algorithms. However, refining the manuscript by addressing the following may be helpful:

 1.       The manuscript could benefit from a more pointed delineation of the specific research gap the authors aim to bridge. How does SLAV-Sim distinguish itself from prevailing simulators in terms of unique features and functionalities?

2.       The choice of raycasting over cameras for observations warrants further explanation. While computational efficiency is highlighted as the rationale, empirical evidence to validate this stance is absent. A deeper exploration discussing the pros and cons of raycasting vis-a-vis cameras would bolster this section's credibility.

3.       Although the overarching methodology appears sound, there exists a noticeable paucity of detailed information concerning simulation setups, employed training hyperparameters, neural network architectures, and other related facets. Such details are indispensable for ensuring reproducibility and allowing readers to fathom the nuances of the proposed implementation.

4.       To comprehensively assess the efficacy of the proposed framework, additional analysis concentrating on simulated driving behaviors, collision frequency, and other pertinent metrics is crucial. Incorporating visual aids like videos or graphic visualizations showcasing the learned driving policies would augment the reader's understanding.

5.       There is some repetition across sections that could be condensed.

6.       References need to be formatted inconsistently throughout.

7.       Expand acronyms before using them (e.g. OEM).

The narrative, while mostly clear, occasionally meanders into repetitive territory. A concise and streamlined presentation would enhance readability.

Reviewer 2 Report

This study revisited existing AV simulation tools and addressed the need for a more accessible solution. After reading it, I have the following comments.

1. The authors said they developed a lightweight simulator. Could the authors show some quantified comparisons to demonstrate why SLAV-Sim is a lightweight simulation platform for AVs.

2. How to describe V2X interactions in SLAV-Sim?

3. How to load real-world human-driven traffic demand (background traffic flows) for testing AVs in SLAV-Sim?

4. How to build car-following and lane-changing behavior of vehicles in SLAV-Sim?

5. In some AV training cases, the platform should tell the optimal decisions for AVs. How does SLAV-Sim calculate the optimal decisions for AVs?

6. Which Unity packages does SLAV-Sim integrate? And what are the responsibilities of Python and Unity in SLAV-Sim?

Round 2

Reviewer 2 Report

1. The "lightweight" is the feature of Unity game engine rather than the originality of SLAV-sim. SLAV-sim just integrates some Unity packages as a platform. The authors should state the contributions of the self-developed methods(i.e., models, algorithm) to "lightweight" .

2. How to ensure that the simulated V2X scenarios in SLAV-sim are equivalent to the real-world scenarios?

3. How to improve the speed of the SLAV-sim simulation process to meet CAV test requirements?

Author Response

Kind Regards
